# Virulence Factors and Antimicrobial Resistance Profile of *Escherichia Coli* Isolated from Laying Hens in Italy

**DOI:** 10.3390/ani12141812

**Published:** 2022-07-15

**Authors:** Lorenzo Gambi, Rachele Rossini, Maria Luisa Menandro, Giovanni Franzo, Francesco Valentini, Giovanni Tosi, Mario D’Incau, Laura Fiorentini

**Affiliations:** 1Istituto Zooprofilattico Sperimentale Della Lombardia e dell’Emilia Romagna (IZSLER), 47122 Forlì, Italy; rachele.rossini@izsler.it (R.R.); francesco.valentini@hotmail.com (F.V.); giovanni.tosi@izsler.it (G.T.); laura.fiorentini@izsler.it (L.F.); 2Reparto Produzione Primaria, Istituto Zooprofilattico Sperimentale Della Lombardia e dell’Emilia Romagna (IZSLER), 25124 Brescia, Italy; 3Dipartimento di Medicina Animale, Produzioni e Salute, Università Degli Studi di Padova, 35020 Legnaro, Italy; marialuisa.menandro@unipd.it (M.L.M.); giovanni.franzo@unipd.it (G.F.); 4Laboratorio Batteriologia Specializzata, Reparto Tecnologie Biologiche Applicate, Istituto Zooprofilattico Sperimentale Della Lombardia e dell’Emilia Romagna (IZSLER), 25124 Brescia, Italy; mario.dincau@izsler.it

**Keywords:** avian pathogenic *Escherichia coli*, APEC, laying hens, colibacillosis, MDR

## Abstract

**Simple Summary:**

Colibacillosis is a disease of great importance in the poultry industry, but many of its features and characteristics still need to be identified. This survey on avian *Escherichia coli* investigated the correlation between the presence of specific virulence genes, antimicrobial resistance features and serogroups. The results highlighted that half of the tested strains were avian pathogenic *Escherichia coli* (APEC). Moreover, a high prevalence of two specific serogroups was detected, namely, O2 and O88. Finally, antimicrobial resistance was lower than in other studies. Further investigations of APEC strains’ antimicrobial resistance features would support farmers, veterinarians and local authorities in planning actions for a better control of colibacillosis in poultry production.

**Abstract:**

Colibacillosis is the most common bacterial disease in the poultry industry. The isolation of *Escherichia coli* (*E. coli*) strains with multiple resistance to various classes of antimicrobials has been increasing in recent years. In this study, antimicrobial resistance features, serotyping and the presence of avian pathogenic *Escherichia coli* (APEC) virulence genes were investigated on a total of 71 *E. coli* strains isolated during outbreaks of colibacillosis in laying hens. The correlation between these features was evaluated. The most frequently isolated serogroups were O2 and O88. Resistance was often detected with nalidixic acid (49%) and ampicillin (38%), while all strains were sensitive to ceftiofur and florfenicol. Overall, 25% of the isolates showed resistance to at least three or more antimicrobial classes (multidrug-resistant strains), and 56% of the isolates were defined as APEC strains (due to the presence of at least five virulence genes). Correlation between the different parameters (virulence genes, serogroup and antimicrobial resistance) did not reveal relevant associations. The comparison of the obtained results with those of similar studies highlighted the importance of continuous monitoring in order to have a better understanding of colibacillosis. An evaluation of the national epidemiological situation would allow, especially with regard to antimicrobial resistance, to focus on the right measures in order to prioritize the available resources for effective disease control.

## 1. Introduction

Colibacillosis is a systemic or localized infection mainly caused by avian pathogenic *Escherichia coli* (APEC), which belongs to the extraintestinal pathogenic *E. coli* (ExPEC) group. Serotypes are classified according to three major antigenic structures: the somatic (O) antigen, flagellar (H) antigen and capsular (K) antigen. The O antigen is particularly important, as it is a portion of lipopolysaccharide, an endotoxin released when the cell undergoes lysis [1].

APEC strain pathogenicity is also related to other virulence factors: adhesins encoded by a temperature-sensitive hemagglutinin gene (tsh), protectins encoded by an increased serum survival gene (iss), toxins encoded by an enteroaggregative toxin gene (astA), vacuolating autotransporter toxin gene (vat), colicin V plasmid operon genes (cvi/cva), iron acquisition systems encoded by an iron-repressible protein gene (irp2) and aerobactin biosynthesis protein gene (iucD). These genes are located in different structures, such as transposons, plasmids, bacteriophages and pathogenicity islands, which may occur individually or in groups [2].

APEC strains may cause one or more of the following diseases: colisepticemia, swollen-head syndrome, peritonitis, salpingitis, orchitis, coligranuloma (Hjarre’s disease), air sac disease, coliform cellulitis (inflammatory process), enteritis, osteomyelitis/synovitis (turkey osteomyelitis complex), panophthalmitis and omphalitis/yolk sac infection [1]. Colibacillosis is the most frequent bacterial disease in poultry, leading to great economic losses and particularly affecting laying hens and all long-term production poultry systems [1,3]. The public health significance of avian *E. coli* is primarily related to antibiotic resistance events, such as carry over and cross resistance [4]. Moreover, zoonotic potential has been identified in some ExPEC serogroups, i.e., Shiga toxin-producing *E. coli* O157 [5,6]. Disease control is performed through antimicrobial treatments and appropriate farm management, along with novel approaches such as the use of probiotics and bacteriophages, aiming to reduce predisposing factors [1,3]. However, the increase in multidrug-resistant (MDR) *E. coli* strains circulating leads to the failure of treatments [7,8]. To date, antimicrobial resistance has been a global menace to human and animal health. To reduce this phenomenon, a one-health approach is needed [9].

APEC strains have been associated with specific serogroups, namely, O1, O2 and O78 [10]. Other serogroups isolated during colibacillosis outbreaks commonly belong to O8, O15, O18, O35, O88, O109 and O115 [11]. In particular, a survey reported that half of the European strains belong to six different serogroups (O1, O2, O5, O8, O18 and O78) [12]. A study carried out in France identified 12 different serogroups, with the most frequently isolated ones being serogroups O2, O8, O25 and O78 [13]; likewise, a Greek survey mainly identified O78, O2 and O11 [14]. Studies from different Asian countries highlight a high prevalence of serogroups O78 and O2 [15,16], but also O1 and O18 displayed a relevant prevalence [17].

Currently, few studies have investigated the distribution of APEC virulence genes in different serogroups and the prevalence of antimicrobial resistance in APEC strains in Italy [7,18]. A deeper insight on serogroups, antimicrobial resistance and specific virulence genes may help to better understand colibacillosis. This study aims to evaluate the features of antimicrobial resistance, serogroups and virulence genes in *E. coli* strains isolated during colibacillosis in industrially reared laying hens in Italy.

## 2. Materials and Methods

### 2.1. Sampling

The study was performed on 71 different *E. coli* strains isolated between 2018 and 2020, originating from as many disease cases as possible in 32 different layer farms located in 10 Italian regions. Half of the farms were placed in 4 provinces of the Emilia Romagna region and supplied 75% of the strains, while the remaining were distributed across 10 more Italian regions. The two-year span allowed to sample some farms more than once, as layer groups changed every 80–90 weeks. No farm performed vaccinations against *E. coli* during the period of the study. Organ sampling was performed either in the field by farm veterinarians, or during necropsy. Whole carcasses or organ specimens were delivered to the laboratory for microbiological examination at refrigeration temperature.

Organ sampling was performed based on any gross lesions related to colibacillosis identified during necropsy.

### 2.2. E. coli Isolation and Identification

Organ surfaces were first sterilized with Bunsen burner flame, followed by carving to expose organ’s solid part or cavity. Material was collected with a sterile inoculation loop or needle and directly inoculated on blood agar plates, whereas Hektoen enteric agar was used for *Enterobacteriaceae* selective growth. Both kinds of plates were then incubated at 37 °C for 24–48 h under aerobic conditions. No enrichment of either organs or bacterial cultures was performed. Culture media were produced according to ISO 11133:2014 by laboratories of Istituto Zooprofilattico Sperimentale della Lombardia ed Emilia Romagna (IZSLER, Brescia, IT). The identification of isolated strains was performed by using Microgen^TM^ GnA+B-iD System kit (Microgen Bioproducts Ltd., Camberley, UK).

### 2.3. Somatic Antigen Identification

All isolated *E. coli* samples were tested for somatic antigen identification through a seroagglutination test. This method allowed the identification of the following 30 different somatic antigens by using the slow agglutination method with agglutinating antisera: O1, O2, O5, O8, O9, O15, O18, O20, O22, O26, O45, O49, O55, O64, O78, O86, O88, O101, O103, O111, O113, O118, O128, O138, O139, O141, O147, O149, O153 and O157. All antisera were purchased from Statens Serum Institut (København, DK). When a strain tested positive for a single antiserum, it was considered as part of the correspondent serogroup. If 2 to 4 antisera tested positive, the choice fell on the one that agglutinated at the standard titer or the lowest dilution. When 5 or more agglutinating antisera tested positive, the reaction was considered autoagglutination and the strain serogroup could not be determined (ND) [19,20,21,22].

### 2.4. Antimicrobial Susceptibility Testing

Antimicrobial susceptibility of the 71 *E. coli* strains was performed according to the Kirby–Bauer disk diffusion method. The following panel of antimicrobials was tested on 69 strains: nalidixic acid (NA, 30 µg), aminosidine (AN, 60 µg), amoxicillin/clavulanic acid association (AMC, 10/20 µg), ampicillin (AMP, 10 µg), apramycin (APR, 15 µg), cephalothin (KF, 30 µg), ceftiofur (EFT, 30 µg), enrofloxacin (ENR, 5 µg), florfenicol (FFC, 30 µg), gentamicin (CN, 10 µg), kanamycin (K, 30 µg), tetracycline (TE, 30 µg) and sulfamethoxazole/trimethoprim association (SXT, 1.25/23.75 µg). The remaining 2 strains were not tested for sulfamethoxazole/trimethoprim. Briefly, isolated E. coli was suspended in sterile saline to reach a final turbidity corresponding to the 0.5 McFarland standard. Afterward, a sterile swab was used to collect the broth culture and streaked once on a Mueller Hinton agar plate. Incubation was performed at 37 °C for 24 h under aerobic conditions. The interpretation of the zone of inhibition was performed in agreement with the guidelines provided by the National Reference Laboratory for Antimicrobial Resistance (Centro Nazionale di Referenza per l’Antibiotico Resistenza, CRAB) and Clinical and Laboratory Standards Institute (CLSI) [23,24,25,26,27,28].

### 2.5. APEC Genes Detection

All 71 isolates were screened for the presence of eight virulence genes (astA, iss, irp2, papC, cvi/cva, iucD, tsh and vat). The detection of APEC virulence genes was performed through a traditional PCR by means of Kylt^®^ APEC (AniCon Labor GmbH, DE) commercial kit according to manufacturer instructions. Briefly, a single bacterial colony was picked from pure cultural material with a sterile inoculation loop and suspended in 500 µL of DNA Extraction-Mix II. Then, after preheating the heating block to a temperature of +100 °C, it was incubated for 10–15 min at 100 ± 3 °C. The sample was, therefore, centrifuged at 10000 g for 5 min, and the supernatant with the DNA extract was immediately used for the PCR procedure. Short-term to long-term storage of DNA extracts could be possible at refrigeration temperature for a few hours or at −20 °C, respectively. The preparation of the master mix consisted of 10 µL of 2× PCR mix, 2 µL of 10 loading dye and 6 µL of primer mix for a total of 18 µL per reaction. Then, for each sample, 2 µL of DNA extract was added. Moreover, two test tubes were also prepared by adding either 2 µL of the positive control or 2 µL of the negative control to the master mix. Reactions were performed at the following conditions: 94 °C for 2 min, followed by 30 cycles of 94 °C/30 s–58 °C/30 s–72 °C/1 min and, lastly, 10 min at 72 °C. The PCR products analysis was conducted under ultraviolet light (UVITEC Cambridge^®^) after electrophoresis on a 2% agarose gel. The interpretation was carried out with Kylt^®^ APEC manual of instructions, as described in Table 1.

### 2.6. Statistical Analysis

Some composite variables were created in addition to raw data, such as APEC strain classification (i.e., those with 5 or more virulence genes) and MDR strains (i.e., resistant to 3 or more antimicrobial classes). Descriptive statistics were estimated to evaluate the distribution and association of all the features displayed by the *E. coli* strains. Correlation between variables (e.g., virulence genes, antimicrobial resistance, serogroup, etc.) was determined by calculating the Goodman–Kruskal tau, a measure of association between categorical variables that were asymmetric, using the Goodman–Kruskal package in R.

## 3. Results

### 3.1. Necropsy

Gross lesions were linked to systemic colibacillosis in 83% (59/71) of the disease cases, featuring aerosacculitis (23%), pericarditis (44%), hepatitis (54%), fibrinous polyserositis (58%), enteritis (34%), fibrinous ovaritis (75%), osteoarthritis and synovitis (8%) and meningitis (9%) during the anatomopathological examination. Only single-organ or multiorgan swabs were available for the remaining 17% (12/71) of the cases, and no description of the lesions was provided.

### 3.2. Somatic Antigen

Seven different serogroups were identified among the 71 strains. The most frequently detected serogroup was O2 (21%), followed by O88 (13%) and O78 (6%). The remaining four serogroups, namely, O1, O9, O4 and O111, were represented by one strain each. Lastly, 55% of the tested strains was not typeable, and was, therefore, addressed as the ND serogroup.

### 3.3. Antimicrobial Susceptibility Testing

Figure 1 and Appendix A show the limited to insignificant resistance to ceftiofur, florfenicol, apramycin and enrofloxacin. In contrast, nalidixic acid, ampicillin and tetracycline were the least efficient antibiotics, with high resistance rates.

MDR strains, i.e., resistant to three or more antimicrobial classes, were 18 out of 71 (25%). Concerning the serogroups and antimicrobial resistance, serogroup O2 (n = 15) displayed a maximum resistance to nalidixic acid (87%) and lower to ampicillin (33%), and only one strain each was resistant to tetracycline and sulfamethoxazole/trimethoprim. Increased exposure sensitivity was rare, with only one strain each for the following antibiotics: enrofloxacin, cephalothin, aminosidine and amoxicillin/clavulanic acid. The results of susceptibility of O2 strains are displayed in Appendix A.

Serogroup O88 (n = 9) displayed lower resistance to antibiotics, with only two out of nine being resistant to ampicillin (22%), and one strain each to tetracycline and aminosidine. Increased exposure sensitivity was similar to that of O2, with only three strains displaying this feature towards only one antibiotic each (ampicillin, cephalothin and kanamycin). In addition, O88 showed full sensitivity towards nalidixic acid, amoxicillin/clavulanic acid, apramycin, ceftiofur, enrofloxacin, florfenicol, gentamicin and sulfamethoxazole/trimethoprim (Appendix A).

### 3.4. Virulence Gene Detection

The PCR analysis showed that few *E. coli* strains displayed the astA gene and papC gene (4/71, 6%, and 10/71, 14%, respectively). Other genes were present as described in Table 2: vat (25/71, 35%), tsh (45/71, 63%), irp2 and cvi/cva (50/71, 70% each), iucD (60/71, 85%) and iss (64/71, 90%).

Kylt^®^ APEC following manufacturer’s instructions identified an *E. coli* strain as APEC if it showed five virulence genes. Following that definition, in this study, 40 of the 71 strains (56%) could be defined as APEC.

Out of 15 serogroup O2 strains, none displayed the astA gene, and only 2 out of 15 displayed the papC gene. On the other hand, all the strains displayed genes iss, irp2 and cvi/cva, 13 out of 15 displayed genes iucD and tsh and 14 out of 15 displayed the vat gene. Of the serogroup O88 strains, 9 out of 71 showed no presence of the astA, papC and vat genes and only one had the irp2 gene. On the contrary, genes iss, cvi/cva, iucD and tsh were always present. The virulence gene distribution between the two serogroups described above was equal or similar for genes astA, iss, cvi/cva, iucD and tsh. Contrarily, genes papC, vat and irp2 were always absent or seldom frequent in O88, while being present in O2 with high frequency (Figure 2). In conclusion, all O2 strains were APEC, while only one out of nine O88 displayed at least five virulence genes.

### 3.5. Serogroups, Virulence Factors and Multidrug Resistance Correlation

The Goodman–Kruskal tau did not highlight any high correlation between serogroups, virulence factors and multidrug resistance. It was possible to point out that a higher correlation of MDR strains were ampicillin-resistant (0.66). No correlation higher than 0.5 was noticed between APEC strains and single virulence genes (Appendix A).

## 4. Discussion

Colibacillosis is the most frequent bacterial disease in the poultry industry, and leads to remarkable economic losses [1,11,29], especially in long farming cycle systems [3].The control of infection should be performed through adequate management strategies aiming at reducing predisposing factors. Still, antimicrobial treatments are the first line of action in cases of infection [1,3]. However, in recent years, a concerning rise in APEC strains featuring multidrug resistance was observed, leading to the reduced efficacy of antimicrobial treatments [7,8]. This study included 71 *E. coli* strains isolated from clinical outbreaks that occurred in 10 Italian regions between 2018 and 2020 in layers.

Somatic antigen characterization allowed for the precise identification of the serogroup in 45% of the strains, with the most recurrent being O2 (21%) and O88 (13%). The first serogroup was one of the most frequently isolated in Europe, along with O1, O5, O8, O18 and O78 [12,13,14], and both O2 and O88 were linked to colibacillosis in avian species [11]. On the other hand, they were less frequent in eastern Asian countries, where the O78 serogroup was dominant [8,15]. In this study, only four strains belonged to the O78 serogroup, with a lower prevalence than expected and described in the literature [2,11,12,15]. The remaining serogroups were O1, O9, O45 and O111. The wide serogroup diversity and prevalence described in the present study and in other surveys is probably linked to peculiarities of the single farms, regions and countries.

In addition to serotyping, a study on the presence of virulence genes was performed with an end-point PCR method. Some authors tried to correlate the number of virulence genes expressed by an *E. coli* strain and the pathogenicity, assuming that the strain must have at least five virulence genes to be listed as APEC [30]. This definition resulted from epidemiological studies about the distribution of various virulence genes in APEC strains and avian faecalis *E. coli* (AFEC), intending to validate a fast biomolecular method to recognize APEC strains [31,32,33]. In this study, the Kylt^®^ APEC commercial kit was used, which allowed for the identification of eight different virulence genes. Based on the definition of APEC given by the kit’s manual of instructions, the results of this study showed a 56% prevalence of APEC strains, higher than the value (31%) highlighted in a study performed in central Italy with the same biomolecular kit [7]. In the present survey, 44% of the strains caused local or systemic colibacillosis despite not being defined as APEC. The most frequently isolated virulence gene was iss (90%), followed by iucD (85%), cvi/cva and irp2 (70%) and tsh (63%). On the other hand, astA was identified only in 6% of the strains, followed by papC (14%) and vat (35%). Gene prevalence was similar to that highlighted by Sgariglia et al. in central Italy [7]. The Iss gene displayed a high prevalence in many other studies, both in APEC and non-APEC strains [15,16,18,34]. The recurrency of Iss might be related to the importance of this specific gene for the pathogenesis of colibacillosis, as it encodes for a lipoprotein that plays a major role in resistance against phagocytosis. On the other hand, the papC gene displayed a low prevalence, as reported in other surveys [18,34]: as the gene has a role in assembling P fimbriae, the results may suggest that death occurred before *E. coli* started producing the pili. The two main serogroups, namely, O2 and O88, displayed a similar distribution of the following virulence genes: astA, iss, cvi/cva, iucD, papC and tsh. On the other hand, the vat gene was recurrent in O2 strains and absent in O88 strains. A higher prevalence of the same gene in O2 APEC strains was also highlighted in a survey from Poland [2]. As a matter of fact, the vat gene, encoding for the vacuolating autotransporter toxin, is rare in nonpathogenic *E. coli* strains [35]. The gene irp2 was identified in all O2 strains, while it was detected only in one O88 strain. At last, all O2 strains and only one O88 strain were eligible to be defined as APEC according to the definition given above.

Antimicrobial susceptibility testing was performed according to the Kirby–Bauer method with a panel of 14 antibiotics. All strains were sensitive to ceftiofur and florfenicol, while the highest percentage of resistance was found with nalidixic acid (49%) and ampicillin (38%). Other studies reported similar results, although with an even higher resistance frequency [7,15,34,36,37,38,39,40]. For example, while this study highlighted a relevant tetracycline resistance (24%), other research showed higher percentages, namely, 30%, 53% and 66% [7,41,42]. The amoxicillin/clavulanic acid association displayed a resistance (13%) similar to the Italian study mentioned above [7], although considerably lower than the value (94%) described in the study by Younis et al. [42], but also higher than Yassin et al. (i.e., 3.3%) [43]. An interesting difference in antimicrobial resistance patterns between O2 and O88 could be observed for nalidixic acid; while O2 strains were frequently resistant (87%), O88 strains did not show resistance to the same molecules. As all O2 strains and only one O88 strain were APEC, resistance to nalidixic acid might be related to one or more of the virulence genes considered in the present survey. Lastly, enrofloxacin resistance was negligible (1%) in the present work, while it was relevant (18%) in the study by Sgariglia et al. (3) and even higher (54%) in a study from China [43].

Our study highlighted a reassuring sensitivity to ceftiofur and enrofloxacin, with no resistance in the first case and only one strain resistant to the second antimicrobial. Both molecules belong to classes listed as critically important antimicrobials (CIAs), namely, third-generation cephalosporins and quinolones, molecules of the highest importance in human medicine [44]. Overall, results about antimicrobial resistance in this work were encouraging compared to data published in 2019 by the European Centre for Disease Prevention and Control (ECDC), reporting an increase in resistance in *E. coli* towards third-generation cephalosporins and quinolones [9]. Moreover, the present study pointed out a lower antimicrobial resistance than in eastern Asian countries [8,34,37,38,39,40,45]. A significant improvement in management and biosecurity measures has featured the Italian poultry farming system in recent years, resulting in a reduction in both antimicrobial use and selective pressure on microorganisms. The higher frequency in the isolation of MDR strains in other studies might be related to different antimicrobial usage policies in the evaluated countries [8,34,37,38,39,40,45]. On the other hand, antimicrobial resistance was similar to data of the Italian study mentioned above for nalidixic acid, amoxicillin/clavulanic acid, ampicillin, gentamicin and kanamycin, while results were different for enrofloxacin, tetracycline and sulfamethoxazole/trimethoprim, as the present study displayed lower percentages of resistance. Nevertheless, the number of MDR strains isolated in the present study (25%) was lower than in central Italy (40%) [7]. This difference could be related to the sampled production system; while our study only consisted of laying system breeds, the survey of central Italy included different systems, breeds and even species, which may have resulted in different uses of antibiotics by farm veterinarians [7]. Many actions can be pursued to decrease antimicrobial use, such as management and biosecurity measures, vaccines, probiotics and bacteriophages [46]. Probiotics in particular have displayed interesting therapeutical and preventive features, i.e., reducing mortality rates in a challenge test with a O78 APEC strain [47].

Based on the obtained results, no obvious correlation was highlighted between features, (i.e., serogroup, MDR and resistance genes). However, further studies based on a broader dataset and accounting for farm characteristics and management procedures may allow for a deeper understanding of the topic.

## 5. Conclusions

The present study highlighted a reassuring, low antimicrobial resistance in the tested strains. Moreover, while all isolated *E. coli* caused colibacillosis-related lesions, only 56% was APEC. The most frequently identified serogroups were O2 and O88, which displayed some differences in the presence of virulence genes and antimicrobial resistance. Further studies of APEC strains should be carried out to assess their evolution over time in terms of antimicrobial resistance and the relationship with management procedures and reduction in antibiotic use. An evaluation of the national epidemiological situation would also help to assess local situations that may differ from neighboring regions or countries. Deeper insight into antimicrobial resistance features should be obtained both with phenotypic tests (e.g., minimum inhibitory concentration) and genotypic assays (e.g., PFGE, next-generation sequencing). Future studies should be performed on farms with comparable farming systems but different management and biosecurity procedures, as we suggested in the present study. Altogether, these goals would help farmers, veterinarians and local authorities to plan actions in line with a one-health approach, in order to control colibacillosis in poultry production and prevent public health emergencies.

## Figures and Tables

**Figure 1 animals-12-01812-f001:**
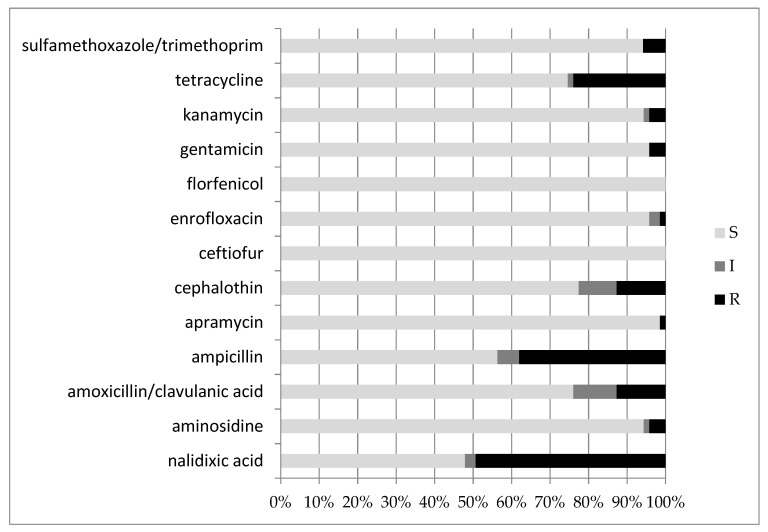
Bar chart reporting the percentages of *E. coli* strains found to be sensitive (S), have increased exposure sensitivity (I) or be resistant (R) to the various antimicrobial molecules tested.

**Figure 2 animals-12-01812-f002:**
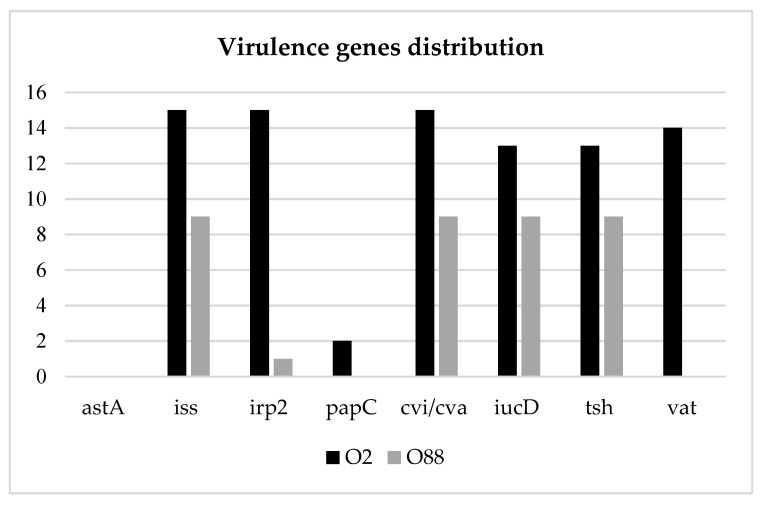
Virulence gene distribution in O2 and O88 *E. coli* strains.

**Table 1 animals-12-01812-t001:** Expected product size for each amplified gene of the positive control.

n. of Band in Positive Control Lane	Amplified Gene	Expected Product Size
1	astA	111 bp
2	iss	309 bp
3	irp2	413 bp
4	pap C	501 bp
5	cvi/cva	598 bp
6	iucD	693 bp
7	tsh	824 bp
8	vat	978 bp

**Table 2 animals-12-01812-t002:** Distribution of virulence genes in *E. coli* strains.

*E. coli* Strains				
Genes	Negative		Positive	
astA	67	94%	4	6%
iss	7	10%	64	90%
irp2	21	30%	50	70%
papC	61	86%	10	14%
cvi/cva	21	30%	50	70%
iucD	11	15%	60	85%
tsh	26	37%	45	63%
vat	46	65%	25	35%

## Data Availability

The data presented in this study are available on request from the corresponding author.

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
