# Peer review of "Virulence Factors and Antimicrobial Resistance Profile of *Escherichia Coli* Isolated from Laying Hens in Italy"

_animals, 2022, doi:10.3390/ani12141812_

Round 1
Reviewer 1 Report
In this study authored by Lorenzo and co-workers the Italian team survey the avian Escherichia coli to investigate the relation between the presence of specific virulence genes, antimicrobial resistance features and serogroup. The anthers highlighted that half of the tested strains were APEC. Furthermore, a high prevalence of two specific serogroups was detected, namely O2 and O88. At last, antimicrobial resistance was lower than in previous studies. The author suggested that more studies on APEC strains’ antimicrobial resistance features to support farmers, veterinarians, and local authorities to actions plan according to a one health approach. This valuable study discuss hot topic regarding antimicrobial resistance and the relationship with management procedures and reduction of antibiotic use. The Ms deserve publication after minor revise. However, I have some comments and suggestions for further improvements of the outcomes of this work in the following sections:
1. The abstract should provide the suggested dose of multispecies probiotic for further investigations.
2. Introduction: Please provide the novelty of this good-well designed work and extended the statement in L 53-54 “Currently, few studies have investigated the prevalence of antimicrobial resistance in APEC strains in Italy [3] to show the add value of this study.
3. Here are some references that may be helpful for extended the literature of economic zoonotic disease in poultry
-Hafez, M. H. and Y. A. Attia (2020). Challenges to the poultry industry: Current perspectives and strategic future after the COVID-19 outbreak. Front. Vet. Sci. doi: 10.3389/fvets.2020.00516,
-Hossain, M.J.; Attia, Y.; Ballah, F.M.; Islam, M.S.; Sobur, M.A.; Islam, M.A.; Ievy, S.; Rahman, A.; Nishiyama, A.; Islam, M.S.; et al. Zoonotic Significance and Antimicrobial Resistance in Salmonella in Poultry in Bangladesh for the Period of 2011–2021. Zoonotic Dis. 2021, 1, 3–24. https://doi.org/ 10.3390/zoonoticdis1010002
-Attia Y.A., Ellakany H.F., Abd El-Hamid A.E., Bovera Fulvia, Ghazaly S.A. (2012). Control of Salmonella enteritidis infection in male layer chickens by acetic acid and/or prebiotics, probiotics and antibiotics. Arch. Geflügelk., 76 (4) 239–245.
4. L 143, Figure 1. bar=Figure 1. Bar
5. L 114 2.6. Statistical analysis and 3.5. Statistical analysis L 189, the statistical statement should be margin in one section, and the title of 2nd statement should be changed to Serogroups, virulence factors, and multidrug resistance.
Reviewer 2 Report
Avian pathogenic Escherichia coli (APEC) is a concerning poultry pathogen. It is one of the main bacteria of commercial and backyard broilers and is widespread throughout the world. APEC strains have several different serovars as well as a diverse repertoire of pathogenic characteristics, which include virulence factors involved in host cell adhesion and invasion, serum resistance factors and toxins. These strains can also carry antimicrobial resistance genes. The manuscript describes the antimicrobial resistance features, serotyping, and presence of APEC virulence genes in 71 E. coli strains isolated during outbreaks of colibacillosis in laying hens from Italy. In my opinion, the research question of the manuscript is very important and it is necessary to publish the main characteristics of APEC strains from different geographic regions and poultry production chains (broilers, layers, breeders).
However the whole manuscript needs much more work before to be ready for a first peer revision. First of all, the title should define at least the geographic region where the samples were collected, since the presented APEC characteristics are very specific and they are not represent APEC from other poultry producing regions of the World. Second the authors have to correct as their names are written (they inverted most of their names). The Simple Summary and the Abstract seem Ok, but I have one important recommendation: to revise the last sentence, since it was not clearly demonstrated the importance of the study in a "One Health approach". I also suggest the authors to highlight the importance of the study for the understanding of the chicken disease (collibacilosis) as well as the consequences for poultry production (more than for an “One Health” approach without a more proper explanation).
In the main body of the manuscript, authors need to improve all sections. The Introduction is very short and does not present up-to-date literature. I would recommend at least three paragraphs to describe the importance of the disease, the main characteristics of APEC strains worldwide, and antimicrobial resistance (before the objectives). It is also necessary to include more recent bibliography (eg, there are more recent articles and books reporting the most common serogroups). Materials and Methods should be more detailed, especially in Sampling procedures. I suggest better describing the characteristics of the farms, as well as the flocks sampled (a map showing the location of the farms would be very welcome). On oppose, it is not necessary to describe that the “Obtained data were organized in an Excel file database”. Please remove it.
I am also very concerned with Results and Discussion. It is difficult to understand some Results as for example the frequency of serogroups. What do the authors mean with “In 55% of those none of the serotypes identifiable by the method used was identified”? There are also many Tables and Figures demonstrating the antimicrobial resistance. The authors need to revise all them and to prepare more informative illustrations. Finally, the Discussion need a total revision and a better structure. Please distribute it in different paragraphs, with a logical organization of the topics. .
Additionally, the text remains with many mistakes and problems in English grammar and style despite the authors have already reviewed once.
Reviewer 3 Report
AUTHORS
Title: Virulence factors and antimicrobial resistance profile of Escherichia coli isolated from laying hens
Reference animals-1805631
Authors have studied antimicrobial resistance features, serotyping, and presence of APEC virulence genes have been investigated on a total of 71 E. coli strains isolated during outbreaks of colibacillosis in laying hens. The manuscript is interesting and has valuable data on an important veterinary pathogen, particularly for poultry. Colibacillosis is the most common bacterial disease in the poultry industry and has a huge impact on economy.
The potential impact on infectious diseases control is high and the manuscript is well-written using robust scientific language. I advise publication after minor modifications are made.
Please define APEC when first used on abstract
Line 33 of abstract don’t start sentence with a numeral
Authors state that correlation between the different parameters (virulence genes, serogroup, and anti-microbial resistance) did not reveal relevant associations. I have however one concern with this has association is measured using other statistical tools (univariate and multivariate logistic regression for example). I would just say that no statistically significant correlation was found.
Line 4 correct “meanace”
Please hyphenate Kirby Bauer
Line 150 do not start with numeral
At the end of results authors report that The Goodman – Kruskal tau did not highlight any correlation between serogroups, virulence factors, and multidrug resistance. I would like to see the correlations and p values here.
Authors also say that it was possible to point out a higher probability of MDR strains to be ampicillin resistant (0,64). Is this the result of the correlation? If so I would tone down the sentence because its not significant. Also please change the comma by a point on 0,64
Again in discussion, authors state that statistical analysis did not highlight a correlation between features (e.g., serogroup, MDR, resistance genes), and differences might be related to farm characteristics and management procedures. I would rephrase this sentence as usually differences are observed using other methods than correlation.
Round 2
Reviewer 2 Report
I have already reviewed the previous version of this submitted article. The entire manuscript has really been improved. Now the title is defining the country where the samples were collected, as APEC characteristics can be very specific to this poultry producing region. The authors also corrected their names! The Abstract has now improved, but I still recommend that authors review the last sentence of the Simple Abstract, as the importance of the study in a "One Health" approach has not been clearly demonstrated. The manuscript highlighted the importance of the study for understanding chicken disease (collibacillosis) as well as the consequences for poultry production. That's quite a lot!
In the main body of the manuscript, the authors have improved all sections. The Introduction now presents updated literature and the paragraphs describe the importance of the disease, the main characteristics of APEC strains worldwide, and antimicrobial resistance. The Materials and Methods are more detailed, mainly in the Sampling procedures. But I still suggest a better description of the farms, highlighting that the samples are mostly from the Emilia Romagna region (considering that the map with the location of the farms will not be shown). The Results section now has more informative Tables and Figures. Discussion has been improved after reviewing other important articles and distributing topics into a more logical organization of paragraphs. Finally, I suggest that authors write a more concise Conclusion section, highlighting the main findings of the present manuscript. And not the need for further studies (these phrases should be transferred to the Discussion).
In addition, the text remains with some errors and problems in English grammar and style.
